# Real-World Evidence of Administration of Biologic Agents in Patients with Severe Asthma: An Analysis of the Respiratory Department of University Hospital of Patras Asthma Registry

**DOI:** 10.3390/jcm14072174

**Published:** 2025-03-22

**Authors:** Ourania Papaioannou, Ioannis Christopoulos, Panagiota Tsiri, Fotios Sampsonas, Kyriakos Karkoulias, Dimosthenis Lykouras, Vasilina Sotiropoulou, Eva Theohari, Dionysios Papalexatos, Dimitrios Komninos, Antonios Christopoulos, Argyrios Tzouvelekis

**Affiliations:** Department of Respiratory Medicine, University Hospital of Patras, 26504 Patras, Greece; ouraniapapaioannou@outlook.com (O.P.); christopoulosi94@gmail.com (I.C.); tsiripanayiota@gmail.com (P.T.); fsampsonas@gmail.com (F.S.); karkoulias@upatras.gr (K.K.); lykouras@upatras.gr (D.L.); v.swtiropoulou@gmail.com (V.S.); eva1733@hotmail.com (E.T.); drpapalex@gmail.com (D.P.); komninos312@gmail.com (D.K.); antonchrist2004@yahoo.gr (A.C.)

**Keywords:** benralizumab, biologic agents, mepolizumab, severe asthma, tezepelumab

## Abstract

**Background:** Real-world data on currently used biologic agents in patients with severe asthma are lacking. **Methods:** In this retrospective study, we recorded between 16 May 2020 and 31 December 2024 consecutive patients who presented to our asthma outpatient clinic received a diagnosis of uncontrolled severe asthma and were treated with biologic agents. Outcomes included a comparison of disease phenotypic characteristics, as well as asthma control, lung function, longitudinal use of corticosteroids, and hospitalizations due to exacerbations at baseline and post-biologic treatment at 6-month follow-up. **Results:** We identified 80 patients with uncontrolled severe asthma treated with biologic agents. The median age (95% CI) at the time of diagnosis was 67.0 (61.0 to 70.0) years. Most patients were female (65.0%, n = 52) and never smokers (51.3%, n = 41). The median value of ACT (95% CI) was 15 (15 to 16) at the time of diagnosis. The mean FEV_1_% predicted ±SD at the baseline was 68.9 ± 22.0. The median value of blood eosinophils (95% CI) was 365 (252 to 448) K/μL in the overall population. One-third (36.3%) of patients were hospitalized due to severe asthma exacerbation in the previous year. Longitudinal use of oral corticosteroids was recorded in 11.3% of included patients. Three patients (3.8%) were treated with omalizumab, 23 patients (28.8%) with mepolizumab, 33 patients (41.2%) with benralizumab and 21 patients (26.2%) with tezepelumab. The median value of ACT (95% CI) post-biologic treatment at 6-month follow-up was 20 (20 to 21), *p* < 0.0001. The mean FEV_1_% predicted ±SD at 6-month follow-up was 77.6 ± 25.2, *p* = 0.12. The median value of blood eosinophils (95% CI) 6 months after initiation of biologic treatment was 100 (40 to 121) K/μL, *p* < 0.0001. Elimination of hospitalizations due to asthma flares was recorded in 97.5% of patients (*p* < 0.0001). With regard to the longitudinal use of oral corticosteroids, we noticed that 96.2% of patients achieved discontinuation. No treatment-related adverse events were noticed. **Conclusions:** The administration of current biologic agents in patients with severe asthma seems to be both effective and safe, sparing the toxicity of oral corticosteroids.

## 1. Introduction

Severe asthma is a heterogeneous, multifactorial disease presented as chronic inflammation of the airways and characterized by several, quite often overlapping phenotypes that correspond to different endotypes [1]. Up to 10% of adults with asthma are characterized as patients with severe asthma, with impaired quality of life and an increased risk of persistent airflow limitation, exacerbations, health care resource use, hospitalizations, and increased mortality [2]. In this subgroup of patients, biologic therapies represent an additional option, especially with regard to minimizing or eliminating exposure to treatment with high-dose inhaled glucocorticoids and oral glucocorticoids. Given the heterogeneity of the severe asthma population, patients need to be thoroughly evaluated to understand the clinical features and outcomes of severe asthma in order to achieve personalized and targeted care [3]. The past few years have seen the advent of six biologic agents targeting various pathways, including anti-immunoglobulin E (IgE) monoclonal antibody for allergic predominant severe asthma with increased IgE, anti-interleukin-5 (IL-5)/anti-IL-5 receptor (R)/anti-IL4R biologics for eosinophilic predominant severe asthma and anti-thymic stromal lymphopoietin (TSLP) monoclonal antibody irrespective of disease endotypes [1,4]. With regards to anti-TSPL tezepelumab, it is important to know that despite the fact that tezepelumab is the first biologic agent with at least some degree of activity in T2 low refractory severe asthma, which remains an unmet need, results from randomized controlled clinical trials suggest that blocking the upstream alarmin TSLP with tezepelumab results in clinically meaningful improvements in asthma control in patients with T2 high asthma. In other words, clinicians could expect a greater response to tezepelumab in those patients with T2 high asthma [5,6]. The choice of biologic agent should include consideration of a variety of clinical and pragmatic factors, including biomarkers such as IgE, eosinophil, and FeNO levels. While large randomized controlled trials have clearly demonstrated the efficacy of biologics in reducing asthma exacerbations, improving quality of life and lung function, and eliminating oral corticosteroids-associated adverse events, therapeutic real-world evidence on current biologic agents in patients with severe asthma, particularly tezepelumab is limited [7,8,9,10,11,12,13,14,15].

This study aims to assess the efficacy and safety profiles of four reimbursed biologic agents (omalizumab, mepolizumab, benralizumab, and tezepelumab) in the everyday clinical setting in a reference center of excellence for asthma management in Western Greece, namely Achaia regional unit, comprising an overall population of 300,000 individuals.

## 2. Methods

### Study Design and Patient Selection

In this retrospective study, we recorded between 16 May 2020 and 31 December 2024 consecutive patients who presented to our asthma outpatient clinic of the Respiratory Department in the University Hospital of Patras, Greece, received a diagnosis of severe asthma and were treated with biologic agents from diagnosis until now. Diagnosis was established following functional examination in the appropriate clinical and laboratory setting. More specifically, we performed spirometry in a microQuark PC-based spirometer from COSMED (OMNIA software, version 2017, Rome, Italy), before and after bronchodilator, to assess lung function and seek objective evidence of variable expiratory airflow limitation (>200 mL and >12% increase in forced expiratory volume in 1 s) [16,17]. Severe asthma was defined as uncontrolled asthma with maximal optimized therapy and treatment of contributory factors or that worsens when high-dose treatment is decreased based on international ERS/ATS guidelines on definition, evaluation, and treatment of severe asthma, as well as GINA recommendations [16,17]. Patients with a follow-up of at least six months were included in the analysis. Data collection and analysis were approved by the Institutional Review Board and the Local Ethics Committee (protocol number 19395/06-Jul-2023). Informed consent was obtained from all individual participants included in the study.

Age, smoking history, asthma control test (ACT), body mass index (BMI), functional indices including forced expiratory volume in 1 s/forced vital capacity (FEV_1_/FVC) and FEV_1_% predicted (pred), blood eosinophils, fractional exhaled nitric oxide (FeNO), immunoglobulin E (IgE), skin prick tests (SPTs), history for previous administration of corticosteroids, hospitalizations due to exacerbations, comorbidities, and treatment modalities applied were recorded.

## 3. Outcome Measures

Outcomes included a comparison of disease phenotypic characteristics, as well as asthma control, lung function, longitudinal use of corticosteroids, and hospitalizations due to exacerbations at baseline and post-biologic treatment at 6-month follow-up.

## 4. Statistical Analysis

With regards to baseline data, summary descriptive statistics were generated with categorical data displayed as absolute numbers and relative frequencies. Continuous data were denoted as mean± standard deviation (SD) or medians with a 95% Confidence Interval (95% CI) following the Kolmogorov–Smirnov test for normality. Mann–Whitney test or t-test was used for the investigation of differences between groups based on the absence or presence of normality. Kruskal–Wallis H test was used to determine if there were statistically significant differences between more groups of an independent variable on an ordinal dependent variable.

## 5. Results

### 5.1. Baseline Patient Characteristics

We included 80 patients with severe asthma treated with biologic agents for at least 6 months. Baseline characteristics are summarized in Table 1. The median age (95% CI) at the time of diagnosis was 67.0 (61.0 to 70.0) years. Most patients were female (65.0%, n = 52) and never smokers (51.3%, n = 41). The mean BMI ± SD was 28.29 ± SD 5.9 kg/m^2^, and the median value of ACT (95% CI) was 15 (15 to 16) at the time of diagnosis. The median value of FEV_1_/FVC (95% CI) was 0.71 (0.69 to 0.73), and the mean FEV_1_% predicted ±SD at the baseline was 68.9 ± 22.0. The median value of blood eosinophils (95% CI) was 365 (252 to 448) K/μL in the overall population. The mean value of FeNO ± SD was 26.3 ± 17.9 ppb, and the median value of IgE (95% CI) was 266 (20 to 1099) IU/mL. Persistent airflow limitation (PAL) phenotype in spirometry was present in 36.3% of patients, while 23.8% of the study population had positive SPTs. Almost one-third of our patients (36.3%, n = 29) were hospitalized due to severe asthma exacerbation in the previous year. Nasal polypoids were present in nine patients (11.3%).

### 5.2. Treatment Modalities Pre-Biologic Agents

Triple inhaled therapy, including inhaled corticosteroids (ICS), long-acting beta agonists (LABAs), and long-acting muscarinic antagonists (LAMAs), was given to the majority of patients (77.5%, n = 62), while 18 patients (22.5%) were given high doses of ICS/LABA. It is notable that all the aforementioned patients who were given high doses of ICS/LABA received inhalers containing extra fine particles. Interestingly, 7.5% of the study population were administered nebulized ICS plus short-acting beta agonists (SABAs) and short-acting muscarinic antagonists (SAMAs), while 3.8% of patients had not received any inhaled therapy pre-triple-inhaled therapy treatment, considering that asthma diagnosis was set following hospitalization due to disease exacerbation. Antileukotrienes plus inhalers were implemented in 10 patients (12.5%). Longitudinal use of oral corticosteroids was recorded in 11.3% of included patients.

### 5.3. Biologic Administration

Three patients (3.8%) were treated with omalizumab (dose and dosing frequency determined by serum total IgE level measured before treatment initiation and by body weight), 23 patients (28.8%) with mepolizumab (100 mg/month), 33 patients (41.2%) with benralizumab (30 mg/4 weeks for the first three doses and subsequently 30 mg/8 weeks) and 21 patients (26.2%) with tezepelumab (210 mg/month). The selection of specific biologic therapy was based on the physician’s decision, taking into account asthma baseline phenotyping (blood eosinophils, FeNO, IgE, SPTs, comorbidities).

### 5.4. Patient Characteristics by Biologic Use

With regards to baseline phenotypic characteristics of patients selected to be treated with mepolizumab, benralizumab, and tezepelumab, subgroup analysis revealed the following: median age (95% CI) at the time of diagnosis was 61.0 (51.0 to 70.7), 65.0 (61.0 to 71.0), and 70.0 (63.2 to 71.0) years for mepolizumab, benralizumab, and tezepelumab groups, respectively. Most patients were never smokers in mepolizumab (52.2%) and benralizumab (63.6%) groups, while the majority of patients administered tezepelumab was ex-smokers (52.4%). Patients treated with tezepelumab exhibited worse lung function tests as indicated by mean FEV1% predicted ±SD at the baseline (69.57 ± 20.62, 70.87 ± 20.62 and 65.53 ± 26.10 in mepolizumab, benralizumab, and tezepelumab groups, respectively, *p* = 0.69). Patients treated with mepolizumab and benralizumab exhibited higher eosinophil counts compared to tezepelumab, as median values of blood eosinophils (95% CI) were 425 (340 to 935) K/μL, 420 (200 to 741) K/μL, and 260 (160 to 370) K/μL in mepolizumab, benralizumab, and tezepelumab groups, respectively (*p* = 0.09). Nasal polypoids as comorbidity were present (21.7%), mostly in the mepolizumab group. A thorough subgroup analysis in asthma baseline phenotyping is summarized in Table 2.

### 5.5. Biologic Efficacy and Safety Profiles

The majority of patients experienced asthma control (ACT > 20) and functional improvement. The median value of ACT (95% CI) post-biologic treatment at 6-month follow-up was 20 (20 to 21), *p* < 0.0001. The median values of ACT pre (baseline)- vs. post (6 months after initiation of biologic agent)-biologic treatment are depicted in Figure 1. The median value of FEV_1_/FVC (95% CI) was 0.74 (0.73 to 0.76), *p* = 0.052, and the mean FEV_1_% predicted ±SD at 6-month follow-up was 77.6 ± 25.2, *p* = 0.12. FEV_1_ and FEV_1_/FVC% tracings pre (baseline)- vs. post (6 months after initiation of biologic agent)-biologic treatment are depicted in Figure 2. The median value of blood eosinophils was lower 6 months after initiation of biologic treatment compared to baseline [365 K/μL (95% CI: 252–448) vs. 100 K/μL (95% CI: 40–121), *p* < 0.0001]. (Figure 3) Elimination of hospitalizations due to asthma exacerbations was recorded in 97.5% of patients (*p* < 0.0001). With regards to longitudinal use of oral corticosteroids, we noticed that 96.2% of patients achieved elimination, while the three patients that remained on maintenance oral corticosteroids—due to concomitant autoimmune disease—achieved reduction in dosing by 25%. No treatment-related adverse events were noticed based on patients’ self-reporting. Characteristics of post-biologic treatment at 6-month follow-up are summarized in Table 3. With regards to the subgroup analysis of tezepelumab, at 6-month follow-up, the median value of FEV1/FVC (95% CI) was 0.75 (0.67 to 0.76), *p* = 0.23, and the mean FEV1% predicted ±SD was 67.0 ± 19.50, *p* = 0.57. Elimination of hospitalizations due to asthma exacerbations, as well as of longitudinal use of oral corticosteroids, were achieved in 100% of patients treated with tezepelumab.

## 6. Discussion

To the best of our knowledge, this was the first study investigating the real-world effectiveness and safety of four different biologic agents in the Greek population with severe asthma. We demonstrated that biologic agents in patients with severe asthma seem to be both effective and safe, leading to asthma control while at the same time sparing the toxicity of oral corticosteroids. Results showed a reduction in oral corticosteroid use along with a reduction in total daily dose compared to baseline. Following 6 months of biologic therapy, only 2.5% of patients experienced hospitalization due to asthma exacerbation. ACT showed disease control, and an acceptable safety profile was also observed. Finally, our study represents one of the first attempts to show asthma remission following the use of tezepelumab in a real-world setting.

Our findings are in line with previous real-life studies showing that omalizumab resulted in significant and sustained improvements in asthma exacerbations, asthma control, and lung function and had a steroid-sparing effect and a good safety profile [18,19]. The international study REALITY-A demonstrated that real-world mepolizumab treatment was clinically effective in patients with severe asthma, resulting in disease control and major reductions in oral corticosteroid use [20]. In accordance with these, based on the RELIght study conducted in the Greek population, mepolizumab demonstrated favorable outcomes in several aspects of disease remission, including the rate of exacerbations, steroid-sparing effect, asthma control, and lung function in patients with severe eosinophilic asthma [21]. Importantly, compelling evidence supports mepolizumab efficacy in both allergic and non-allergic severe asthma phenotypes [22]. Our data are also in line with those of Menzella et al., which showed significant improvements in lung function and asthma control following treatment with benralizumab for severe refractory eosinophilic asthma [23]. In a large group of severe eosinophilic asthmatics, mepolizumab and benralizumab both improved disease parameters, such as reducing exacerbations, improving FEV_1_ and depleting blood eosinophils [24]. As already mentioned, the anti-IL-5 monoclonal antibodies, mepolizumab and benralizumab, have presented a major step forward for patients with eosinophilic severe asthma. Nevertheless, while both treatments act on the same cytokine pathway, their mechanisms of action are quite different. Mepolizumab directly binds to and inactivates circulating IL-5. Benralizumab, on the other hand, binds to the eosinophil cell membrane at the IL-5 receptor, preventing activation of the receptor but simultaneously inducing apoptosis of the eosinophil by cross-binding with natural killer cells [24]. It is important to know that eosinophil levels are differently influenced by biologics depending on their mechanism of action. Data derived from the literature show that eosinophil depletion was significantly greater with benralizumab than mepolizumab [24,25,26].

With regards to tezepelumab, a small-scale real-world study is available demonstrating the clinical improvement associated with tezepelumab treatment in severe uncontrolled asthma, independent of inflammatory biomarkers, eosinophilic profile, or previous biological failure [27]. Moreover, Greig et al. very recently showed that tezepelumab confers significant improvements in small airway function in terms of oscillometry parameters, including FEF25–75% [28]. It is important to know that in our country, Greece, four biologic agents, including omalizumab, mepolizumab, benralizumab, and tezepelumab, are indicated and currently fully reimbursed by national health committees for patients with uncontrolled severe asthma.

PAL phenotype, as assessed by spirometry, was present in 36.3% of our cohort and represents undoubtedly an interesting subgroup of patients with severe asthma. Previous studies on PAL in asthma are scarce, including relatively small populations, mostly restricted to severe asthma, or with no longitudinal data. In a recent post hoc analysis of the ATLANTIS study, Kole et al. revealed that PAL was associated—even in patients with mild asthma—with eosinophilic inflammation and a higher risk of exacerbations [29].

Our study presents some limitations. This registry was not scheduled to provide mechanistic data, and as it was a hospital-based epidemiological study with patient characteristics, the quality of data is characterized by sampling bias. Nevertheless, real-world studies can provide complementary data on treatment effectiveness beyond highly selective randomized controlled trial patient populations and bridge this “efficacy-effectiveness” gap, providing valuable sources of real-world data on asthma characteristics, trends, and treatment outcomes, which can inform improved management strategies. Asthma is, by definition, a heterogeneous disease with overlapping pathogenetic pathways in the majority of patients, and we cannot change this reality in an everyday clinical setting. Importantly, the International Severe Asthma Registry (ISAR), the global registry for adults with severe asthma—the largest repository of real-world data on severe asthma, curating data on nearly 35,000 patients from 28 countries worldwide—clearly revealed that there are overlapping pathogenic pathways based on used biomarkers (IgE, FeNO, blood eosinophils) in 65% of enrolled patients [30,31]. These overlapping endotypes reflect the overlapping phenotypes (allergic eosinophilic, non-allergic eosinophilic, T2 low asthma) that may coexist within the same patient. These data further support our approach to include in our analysis mixed asthma phenotypes, as this is in line not only with real-life clinical cases but also signifies the true heterogeneous nature of the disease per se. With regards to the fact that 18 patients (22.5%) were given high doses of ICS/LABA without LAMA, our intention was not to communicate the message that physicians should not follow the stepwise approach of GINA recommendations but in real-life clinical settings, physicians in selected cases taking into account severity of previous exacerbations, higher levels of T2 inflammation biomarkers and patient’s preferences can apply biologics in earlier steps (Step 4) [30,31]. Based on the bibliography, 30% of patients with severe asthma do not respond to applied treatment; therefore, we believe that our approach is not an uncommon clinical practice and reflects reality and the real pathophysiology of this heterogeneous disease. In our cases, our approach not to follow the recommendations of GINA for stepwise therapeutic escalations in this minority of patients was the correct one based on treatment response criteria, as indicated by disease control, elimination of exacerbations even in the absence of OCS, and finally, based on functional improvement. Finally, conclusions of long-term follow-up assessment—beyond 6 months—were not feasible due to the recent reimbursement of tezepelumab in Greece.

In conclusion, our real-life clinical data support the effectiveness and safety of biologic agents in severe asthma through the elimination of flares, reduction in blood eosinophilia, and improvement of lung function. Future real-world evidence from larger registries is greatly anticipated.

## Figures and Tables

**Figure 1 jcm-14-02174-f001:**
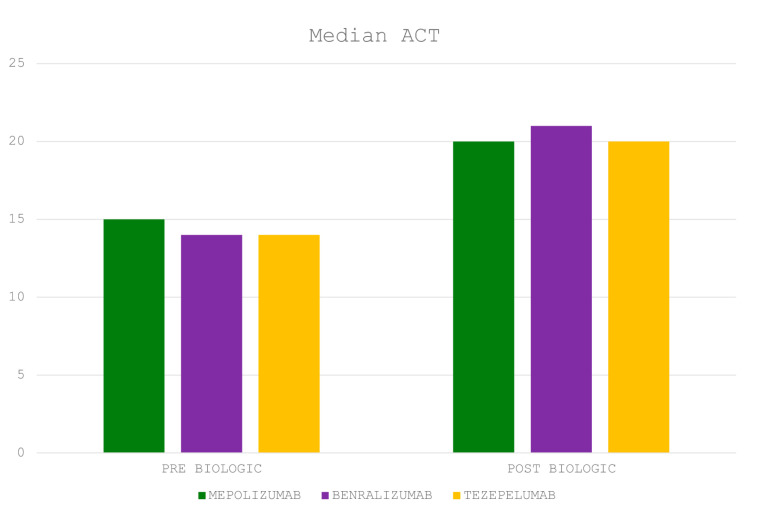
Median values of ACT pre (baseline)- vs. post (6 months after initiation of biologic agent)-biologic treatment (*p* < 0.0001).

**Figure 2 jcm-14-02174-f002:**
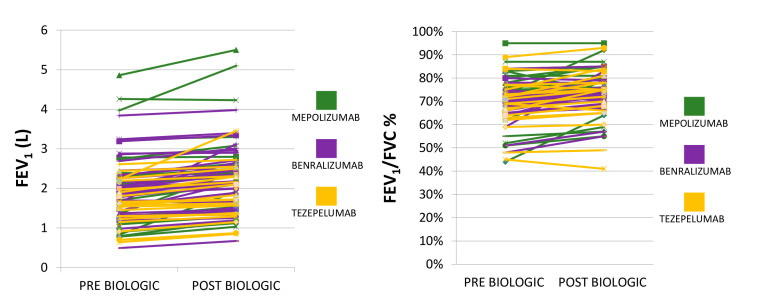
FEV1 and FEV1/FVC% tracings pre (baseline)- vs. post (6 months after initiation of biologic agent)-biologic treatment (*p* = 0.08 and *p* = 0.052, respectively).

**Figure 3 jcm-14-02174-f003:**
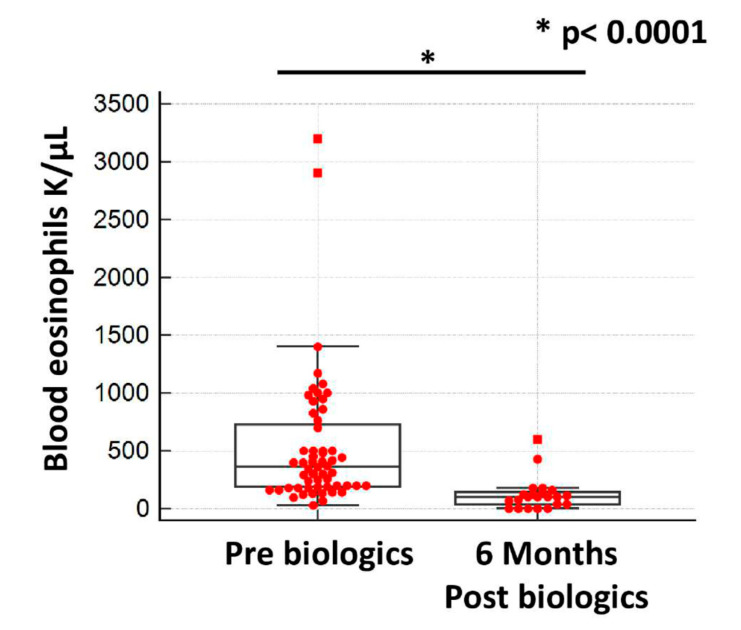
Median value of blood eosinophils was lower 6 months after initiation of biologic treatment compared to baseline [365 K/μL (95% CI: 252–448) vs. 100 K/μL (95% CI: 40–121), *p* < 0.0001].

**Table 1 jcm-14-02174-t001:** Baseline characteristics.

Characteristics	(N, %)
Total number of patients	80
Age (95% CI)	67.0 (61.0 to 70.0)
Male/female	28 (35.0%)/52 (65.0%)
Current smokers/ex-smokers	9 (11.2%)/30 (37.5%)
Never smokers	41 (51.3%)
BMI (kg/m^2^) ± SD	28.29 ± 5.90
ACT (95% CI)	15 (15 to 16)
FEV_1_/FVC (95% CI)	0.71 (0.69 to 0.73)
FEV_1_% predicted ± SD	68.9 ± 22.0
PAL phenotype	29 (36.3%)
Eosinophils (K/μL) (95% CI)	365 (252 to 448)
FeNO (ppb) ± SD	26.3 ± 17.9
IgE (IU/mL) (95% CI)	266 (20 to 1099)
Positive SPTs	19 (23.8%)
Arterial hypertension	29 (36.3%)
Dyslipidemia	27 (33.8%)
Chronic heart disease	17 (21.3%)
Nasal polypoids	9 (11.3%)
GERD	6 (7.5%)
Bronchiectasis	6 (7.5%)
Hospitalization due to exacerbation in the previous year	29 (36.3%)

Abbreviations: ACT: asthma control test, BMI: body mass index, CI: Confidence Interval, FeNO: fractional exhaled nitric oxide, FEV_1_: forced expiratory volume in 1 s, FVC: forced vital capacity, GERD: gastroesophageal reflux disease, IgE: immunoglobulin E, PAL: persistent airflow limitation, SD: standard deviation, SPTs: skin prick tests.

**Table 2 jcm-14-02174-t002:** Subgroup analysis in asthma baseline phenotyping.

Baseline Phenotypic Characteristics	Anti-IL-5 (Mepolizumab) N = 23	Anti-IL-5R (Benralizumab) N = 33	Anti-TSLP (Tezepelumab) N = 21
Age (95% CI)	61.0 (51.0 to 70.7)	65.0 (61.0 to 71.0)	70.0 (63.2 to 71.0)
Male/female	11 (47.8%)/12 (52.2%)	10 (30.3%)/23 (69.7%)	6 (28.6%)/15 (71.4%)
Current smokers/ex-smokers	2 (8.7%)/9 (39.1%)	3 (9.1%)/9 (27.3%)	3 (14.3%)/11 (52.4%)
Never smokers	12 (52.2%)	21 (63.6%)	7 (33.3%)
BMI (kg/m^2^) ± SD	27.02 ± 4.4	28.55 ± 6.6	30.52 ± 5.41
FEV1/FVC (95% CI)	0.73 (0.62 to 0.80)	0.71 (0.67 to 0.73)	0.70 (0.63 to 0.74)
FEV1% predicted ± SD	69.57 ± 20.62	70.87 ± 21.21	65.53 ± 26.10
Eosinophils (K/μL) (95% CI)	425 (340 to 935)	420 (200 to 741)	260 (160 to 370)
Nasal polypoids	5 (21.7%)	1 (3.0%)	1 (4.8%)
Positive SPTs	4 (17.4%)	5 (15.2%)	4 (19.0%)
Hospitalization/exacerbation in the previous year	15 (65.2%)	9 (27.3%)	4 (19.0%)
Oral corticosteroids (longitudinal use)	5 (21.7%)	2 (6.0%)	2 (9.5%)

Abbreviations: BMI: body mass index, CI: Confidence Interval, FEV_1_: forced expiratory volume in 1 s, FVC: forced vital capacity, SD: standard deviation, SPTs: skin prick tests.

**Table 3 jcm-14-02174-t003:** Characteristics post-biologic treatment (6-month follow-up).

Characteristics	(N, %)
ACT (95% CI)	20 (20 to 21)
FEV1/FVC (95% CI)	0.74 (0.73 to 0.76)
FEV1% predicted ± SD	77.6 ± 25.2
Eosinophils (K/μL) (95% CI)	100 (40 to 121)
Severe exacerbations/Hospitalizations	2/80 (2.5%)
Oral corticosteroids (longitudinal use 7.5 mg/day prednisone-concomitant autoimmune disease)	3/80 (3.8%)

Abbreviations: ACT: asthma control test, CI: Confidence Interval, FEV1: forced expiratory volume in 1 s, FVC: forced vital capacity, SD: standard deviation.

## Data Availability

Data are available on request. OP and AT have full access to the data and are the guarantors for these data.

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
