# Peer review of "Real-World Evidence of Administration of Biologic Agents in Patients with Severe Asthma: An Analysis of the Respiratory Department of University Hospital of Patras Asthma Registry"

_jcm, 2025, doi:10.3390/jcm14072174_

Round 1

Reviewer 1 Report (New Reviewer)

Comments and Suggestions for Authors

This is an interesting, properly designed and elegantly presented real-world study of the efficacy of various biologics in the treatment of severe asthma in greek patients. The topic is not of high novelty, however, it nicely falls into current trends of acquiring real-world evidence of treatment efficacy.

The study population is well described in terms of demographics and clinical characteristics. Results are clearly and comprehensively presented and I have no issues to be raised with this regard.

Below please find minor comments:

Eosinophils levels are differently influenced by biologics depending on their mechanism of action – this should be addressed in the discussion.

Do the Authors have any thoughts, comments or further plans regarding efficacy of biologics in the PAL phenotype subgroup? The PAL group has been analyzed in the context of triple therapy effectiveness, this would be interesting to have real-practice data on biologics in them. I encourage the Authors to provide some comments, at their discretion, whatsoever.

Page 3 line 11: this sentence is not quite clear and seems incongruous and superfluous – there is no further context of mentioning the whole cohort in my opinion.

Author Response

Thank you for your valuable comments and the fast peer-review process. Comments obviously stem from experts in the field and therefore we are grateful, as they really helped us to improve the quality of our manuscript. 

Our point-by-point reply to reviewers follows:

Reviewer: 1

This is an interesting, properly designed and elegantly presented real-world study of the efficacy of various biologics in the treatment of severe asthma in Greek patients. The topic is not of high novelty; however, it nicely falls into current trends of acquiring real-world evidence of treatment efficacy.

The study population is well described in terms of demographics and clinical characteristics. Results are clearly and comprehensively presented and I have no issues to be raised with this regard.

Below please find minor comments:

Eosinophils levels are differently influenced by biologics depending on their mechanism of action – this should be addressed in the discussion.

AU: Thank you for your comment. We added the following in discussion section:

“As already mentioned, the anti-IL-5 monoclonal antibodies, mepolizumab and benralizumab, have presented a major step forward for patients with eosinophilic severe asthma. Nevertheless, while both treatments act on the same cytokine pathway, their mechanisms of action are quite different. Mepolizumab directly binds to, and inactivates, circulating IL-5. Benralizumab, on the other hand, binds to the eosinophil cell membrane at the IL-5 receptor, preventing activation of the receptor, but simultaneously inducing apoptosis of the eosinophil by cross-binding with natural killer cells (24). It is important to know that eosinophils levels are differently influenced by biologics depending on their mechanism of action. Data derived from the literature show that eosinophil depletion was significantly greater with benralizumab than mepolizumab (24-26).”

Do the Authors have any thoughts, comments or further plans regarding efficacy of biologics in the PAL phenotype subgroup? The PAL group has been analyzed in the context of triple therapy effectiveness, this would be interesting to have real-practice data on biologics in them. I encourage the Authors to provide some comments, at their discretion, whatsoever.

AU: Thank you for your comment. PAL phenotype is undoubtedly an interesting subgroup of patients with severe asthma. We added the following in discussion section:

“PAL phenotype as assessed by spirometry was present in 36.3% of our cohort and represents undoubtedly an interesting subgroup of patients with severe asthma. Previous studies on PAL in asthma are scarce, including relatively small populations, mostly restricted to severe asthma, or with no longitudinal data. In a recent post-hoc analysis of the ATLANTIS study, Kole at al revealed that PAL was associated -even in patients with mild asthma- with eosinophilic inflammation and a higher risk of exacerbations (29).”

Page 3 line 11: this sentence is not quite clear and seems incongruous and superfluous – there is no further context of mentioning the whole cohort in my opinion.

AU: Thank you for your comment. We agree that it may be incongruous and superfluous, so we erased this sentence.  It had been added in order to show that the percentage of severe asthma patients in our cohort was 5.3% (80/1517) almost identical to that reported in current literature.

Reviewer 2 Report (New Reviewer)

Comments and Suggestions for Authors

My comments:

                In general, this is an interesting research. The authors aim to assess efficacy and safety profiles of four reimbursed biologic agents (omalizumab, mepolizumab, benralizumab and tezepelumab) in the everyday clinical setting in a reference center of excellence for asthma management in Western Greece. They found that administration of current biologic agents in patients with severe asthma seems to be both effective and safe sparing toxicity of oral corticosteroids. However, there are some points should be improved.

Majors

                You mentioned in the introduction that “While large randomized controlled trials have clearly demonstrated efficacy of biologics in reducing asthma exacerbations, improving quality of life and lung function and eliminating oral corticosteroids-associated adverse events; yet therapeutic real-world evidence on current biologic agents in patients with severe asthma, particularly tezepelumab is limited.”. Moreover you also mentioned in the discussion that “Finally, our study represents one of the first attempts to show asthma remission following use of tezepelumab in the real-world setting” and “Finally, conclusions of long-term follow-up assessment -beyond 6 months- were not feasible, due to recent reimbursement of tezepelumab in Greece.” However, the subgroup analysis of tezepelumab was not presented in the results section. Thus, the subgroup analysis of tezepelumab on asthma exacerbations, quality of life, lung function, eliminating oral corticosteroids-associated adverse events, and hospitalization should be added.

  1. Abstract

                - Please check a number and percentage of the following sentence “Most patients 18 were female (52.0%, n=65)…..”

  1. Materials and Methods

                - In the study design section. You stated that “More specifically, we performed spirometry in microQuark PC-based spirometer from COS-MED, before and after bronchodilator, to assess lung function and seek objective evidence of variable expiratory airflow limitation (>200mL and >12% increase in forced expiratory volume in 1 second).” Please provide the reference citation.

  1. Results

                - Please check a number and percentage of the following sentence “Most patients 18 were female (52.0%, n=65)…..”

                - Many duplicates in text and in the table. Thus, the results section should be revised.

                - In figure 1 and in sentence “Median values of ACT pre (baseline) vs post (6 months after initiation of biologic agent) biologic treatment are depicted in Figure 1.” The p-value should be provided.

                - In figure 2 and in sentence “Median value of FEV1/FVC (95% CI) was 0.74 (0.73 to 0.76) and mean FEV1% predicted ±SD at 6-month follow-up was  77.6 ± 25.2. FEV1 and FEV1/FVC% tracings pre (baseline) vs post (6 months after initiation of biologic agent) biologic treatment are depicted in Figure 2.” The p-value should be provided.

                - The percentage of hospitalization should be compared between pre and post treatment, the p-value should be provided.

                - In Table 1, the distribution of FeNO should be re-checked, and the results of FeNO should be changed accordingly.

Author Response

Thank you for your valuable comments and the fast peer-review process. Comments obviously stem from experts in the field and therefore we are grateful, as they really helped us to improve the quality of our manuscript. 

Our point-by-point reply to reviewers follows:

Reviewer: 2

In general, this is an interesting research. The authors aim to assess efficacy and safety profiles of four reimbursed biologic agents (omalizumab, mepolizumab, benralizumab and tezepelumab) in the everyday clinical setting in a reference center of excellence for asthma management in Western Greece. They found that administration of current biologic agents in patients with severe asthma seems to be both effective and safe sparing toxicity of oral corticosteroids. However, there are some points should be improved.

Majors

                You mentioned in the introduction that “While large randomized controlled trials have clearly demonstrated efficacy of biologics in reducing asthma exacerbations, improving quality of life and lung function and eliminating oral corticosteroids-associated adverse events; yet therapeutic real-world evidence on current biologic agents in patients with severe asthma, particularly tezepelumab is limited.”. Moreover you also mentioned in the discussion that “Finally, our study represents one of the first attempts to show asthma remission following use of tezepelumab in the real-world setting” and “Finally, conclusions of long-term follow-up assessment -beyond 6 months- were not feasible, due to recent reimbursement of tezepelumab in Greece.” However, the subgroup analysis of tezepelumab was not presented in the results section. Thus, the subgroup analysis of tezepelumab on asthma exacerbations, quality of life, lung function, eliminating oral corticosteroids-associated adverse events, and hospitalization should be added.

AU: Thank you for your comment. We added the following in results section:

“With regards to subgroup analysis of tezepelumab, at 6-month follow-up, median value of FEV1/FVC (95% CI) was 0.75 (0.67 to 0.76), p=0.23 and mean FEV1% predicted ±SD was 67.0 ± 19.50, p=0.57. Elimination of hospitalizations due to asthma exacerbations, as well as of longitudinal use of oral corticosteroids were achieved in 100% of patients treated with tezepelumab.”

Median values of ACT pre (baseline) vs post (6 months after initiation of tezepelumab) are depicted in Figure 1.

Abstract

                - Please check a number and percentage of the following sentence “Most patients  were female (52.0%, n=65)…..”

AU: Thank you for your comment. We would like to apologize for this. We corrected it.

“(65.0%, n=52)”

Materials and Methods

                - In the study design section. You stated that “More specifically, we performed spirometry in microQuark PC-based spirometer from COS-MED, before and after bronchodilator, to assess lung function and seek objective evidence of variable expiratory airflow limitation (>200mL and >12% increase in forced expiratory volume in 1 second).” Please provide the reference citation.

AU: Thank you for your comment. We provided reference citations.

“Chung KF, Wenzel SE, Brozek JL, Bush A, Castro M, Sterk PJ, et al. International ERS/ATS guidelines on definition, evaluation and treatment of severe asthma. Eur Respir J. 2014;43(2):343-73.”

2024 GINA SEVERE ASTHMA [Available from: https://ginasthma.org/severe-asthma]

Results

                - Please check a number and percentage of the following sentence “Most patients were female (52.0%, n=65)…..”

AU: Thank you for your comment. We would like to apologize for this. We corrected it.

“(65.0%, n=52)”

                - Many duplicates in text and in the table. Thus, the results section should be revised.

AU: Thank you for your comment. We revised results section, as requested. Selected sentences were erased in order to avoid duplicates.

                - In figure 1 and in sentence “Median values of ACT pre (baseline) vs post (6 months after initiation of biologic agent) biologic treatment are depicted in Figure 1.” The p-value should be provided.

AU: Thank you for your comment. P-value was <0.0001. We provided in Figure 1.

                - In figure 2 and in sentence “Median value of FEV1/FVC (95% CI) was 0.74 (0.73 to 0.76) and mean FEV1% predicted ±SD at 6-month follow-up was 77.6 ± 25.2. FEV1 and FEV1/FVC% tracings pre (baseline) vs post (6 months after initiation of biologic agent) biologic treatment are depicted in Figure 2.” The p-value should be provided.

AU: Thank you for your comment. P-value was 0.08 for FEV1 and 0.052 for FEV1/FVC, respectively. They were provided in Figure 2.

                - The percentage of hospitalization should be compared between pre and post treatment, the p-value should be provided.

AU: Thank you for your comment. We added p-value that was <0.0001.

                - In Table 1, the distribution of FeNO should be re-checked, and the results of FeNO should be changed accordingly.

AU: Thank you for your comment. We rechecked distribution of FeNO and it was correct.

Round 2

Reviewer 2 Report (New Reviewer)

Comments and Suggestions for Authors

All of my comments have been addressed by authors. This manuscript can be accepted for publication. 

This manuscript is a resubmission of an earlier submission. The following is a list of the peer review reports and author responses from that submission.

Round 1

Reviewer 1 Report

Comments and Suggestions for Authors "The past few years have seen the advent of six 48 biologic agents targeting various pathways including anti-immunoglobulin E (IgE) mon- 49 oclonal antibody for allergic predominant severe asthma with increased IgE, anti-inter- 50 leukin-5 (IL-5)/anti-IL-5 receptor (R)/anti-IL4R biologics for eosinophilic predominant se- 51 vere asthma and anti-thymic stromal lymphopoietin (TSLP) monoclonal antibody irre- 52 spective of disease endotypes. (1, 4)" It should be mentioned here that anti-TSLP therapy is much more effective in patients with type 2 high asthma versus type 2 low asthma. (PMID: 35131510.)   "Triple inhaled therapy including inhaled corticosteroids (ICS), long-acting beta ago- 119 nists (LABAs) and long‐acting muscarinic antagonists (LAMAs) were given to the major- 120 ity of patients (66.2%, n=53), while 18 patients (22.5%) were given high-doses of 121 ICS/LABA." Do we know if any of the inhalers contained extra fine particles?   "With regards to tezepelumab, only a small-scale real-world study is avail- 209 able demonstrating the clinical improvement associated with tezepelumab treatment in 210 severe uncontrolled asthma, independent of inflammatory biomarkers, eosinophilic pro- 211 file, or previous biological failure." It might be worth discussing this recent article (PMID: 39870209) looking at the effect of tezepelumab on small airways. Did you have any data on FEF25-75%? Comments on the Quality of English Language

could be improved in terms of grammar and syntax

Author Response

Thank you for your valuable comments and the fast peer-review process. Comments obviously stem from experts in the field and therefore we are grateful, as they really helped us to improve the quality of our manuscript.  

Our point-by-point reply to reviewers follows:
Reviewer: 1
"The past few years have seen the advent of six biologic agents targeting various pathways including anti-immunoglobulin E (IgE) monoclonal antibody for allergic predominant severe asthma with increased IgE, anti-interleukin-5 (IL-5)/anti-IL-5 receptor (R)/anti-IL4R biologics for eosinophilic predominant severe asthma and anti-thymic stromal lymphopoietin (TSLP) monoclonal antibody irrespective of disease endotypes. (1, 4)" It should be mentioned here that anti-TSLP therapy is much more effective in patients with type 2 high asthma versus type 2 low asthma. (PMID: 35131510.)   "Triple inhaled therapy including inhaled corticosteroids (ICS), long-acting beta agonists (LABAs) and long‐acting muscarinic antagonists (LAMAs) were given to the majority of patients (66.2%, n=53), while 18 patients (22.5%) were given high-doses of ICS/LABA." Do we know if any of the inhalers contained extra fine particles?   "With regards to tezepelumab, only a small-scale real-world study is available demonstrating the clinical improvement associated with tezepelumab treatment in severe uncontrolled asthma, independent of inflammatory biomarkers, eosinophilic profile, or previous biological failure." It might be worth discussing this recent article (PMID: 39870209) looking at the effect of tezepelumab on small airways. Did you have any data on FEF25-75%?
1) It should be mentioned here that anti-TSLP therapy is much more effective in patients with type 2 high asthma versus type 2 low asthma. (PMID: 35131510.)
AU: Thank you for your comment. We added the following in introduction section: 
“With regards to anti-TSLP tezepelumab, it is important to know that despite the fact that tezepelumab is the first biologic agent with at least some degree of activity in T2 low refractory severe asthma which remains an unmet need, results from randomized controlled clinical trials suggest that blocking the upstream alarmin TSLP with tezepelumab results in clinically meaningful improvements in asthma control in patients with T2 high asthma. In other words, clinicians could expect a greater response to tezepelumab in those patients with T2 high asthma.”
We added relevant references, as well.
2) Do we know if any of the inhalers contained extra fine particles?
AU: Thank you for your comment. We would like to apologize for not reporting that.
We added the following in results section:
“It is notable that all the aforementioned patients that were given high-dose of ICS/LABA received inhalers contained extra fine particles.”
It is important to know that in our country, Greece, high-dose triple inhaled therapy is not available through an inhaler containing extra fine particles.
3) It might be worth discussing this recent article (PMID: 39870209) looking at the effect of tezepelumab on small airways. Did you have any data on FEF25-75%?
AU: Thank you for your comment. With regards to the effect of tezepelumab on small airways we added this in discussion section:
“Moreover, Greig et al, very recently, showed that tezepelumab confers significant improvements in small airway function in terms of oscillometry parameters including FEF25-75%.”
We added relevant reference, as well. Unfortunately, we did not have data on FEF25-75%.

Reviewer 2 Report

Comments and Suggestions for Authors

I have carefully reviewed the manuscript titled "Real-world evidence of administration of biologic agents in patients with severe asthma: an analysis of the Respiratory Department of University Hospital of Patras asthma registry." The study addresses an important and clinically relevant topic regarding the effectiveness and safety of biologic therapies in severe asthma. However, the manuscript presents several critical methodological flaws and inconsistencies that significantly undermine its validity and impact. Due to these substantial limitations, I do not recommend this manuscript for publication in its current form.

Point 1.

The follow-up period of six months is insufficient for assessing long-term efficacy and safety outcomes.

Point 2.

The criteria for asthma diagnosis are not clearly defined, raising concerns about patient selection.

Point 3.

The methodology lacks details on the spirometry instruments used, making it difficult to assess the reliability of lung function measurements.

Point 4. 

The manuscript does not provide clear criteria for selecting specific biologic therapies for individual patients, introducing a risk of selection bias. Moreover, the inclusion of both allergic and non-allergic severe asthma treatments in the same analysis is methodologically flawed and inappropriate.

Point 5. 

Biomarkers such as blood eosinophils and FeNO, although mentioned, are not adequately analyzed in relation to treatment response.

Point 6. 

The manuscript states: "7.5% of the study population were administered nebulized ICS plus short-acting beta agonists (SABAs) and short-acting muscarinic antagonists (SAMAs), while 3.8% of patients had not received any inhaled therapy pre-biologic treatment, considering that asthma diagnosis was set following hospitalization due to disease exacerbation." This raises serious concerns about the appropriateness of biologic therapy in patients who were not on high-dose ICS/LABA prior to treatment. According to GINA 2024, biologic therapy is indicated only for severe asthma that remains uncontrolled despite optimized high-dose ICS-LABA treatment or requires high-dose ICS-LABA to maintain control.

Point 7. 

No adverse events are reported, which raises concerns about potential reporting bias or inadequate safety monitoring.

Point 8. 

The text formatting is inconsistent, particularly between tables and the main body of the manuscript.

The superscript number for affiliations is missing.

The reference list requires review for adherence to journal guidelines, and citation numbers should be placed before the period.

Author Response

Thank you for your valuable comments and the fast peer-review process. Comments obviously stem from experts in the field and therefore we are grateful, as they really helped us to improve the quality of our manuscript.  

Our point-by-point reply to reviewers follows:

Reviewer: 2
I have carefully reviewed the manuscript titled "Real-world evidence of administration of biologic agents in patients with severe asthma: an analysis of the Respiratory Department of University Hospital of Patras asthma registry." The study addresses an important and clinically relevant topic regarding the effectiveness and safety of biologic therapies in severe asthma. However, the manuscript presents several critical methodological flaws and inconsistencies that significantly undermine its validity and impact. Due to these substantial limitations, I do not recommend this manuscript for publication in its current form.
Point 1.
The follow-up period of six months is insufficient for assessing long-term efficacy and safety outcomes.
AU: Thank you for your comment. We strongly agree that 6-month-follow-up is not adequate for assessing long-term efficacy and safety outcomes and we explained it as a limitation of our study: “conclusions of long-term follow-up assessment -beyond 6 months- were not feasible, due to recent reimbursement of tezepelumab in Greece.”
To be more explicit we modified our conclusions as follows:
“Administration of current biologic agents in patients with severe asthma seems to be both effective and safe sparing toxicity of oral corticosteroids.”
“We demonstrated that biologic agents in patients with severe asthma seem to be both effective and safe leading to asthma control while at the same time spare toxicity of oral corticosteroids.”
Point 2.
The criteria for asthma diagnosis are not clearly defined, raising concerns about patient selection.
AU: Thank you for your comment. We clarified diagnostic approach in methods section:
“Diagnosis was established following functional examination in the appropriate clinical and laboratory setting. More specifically, we performed spirometry in microQuark PC-based spirometer from COSMED, before and after bronchodilator, to assess lung function and seek objective evidence of variable expiratory airflow limitation (>200mL and >12% increase in forced expiratory volume in 1 second). Severe asthma was defined as uncontrolled asthma with maximal optimized therapy and treatment of contributory factors or that worsens when high-dose treatment is decreased based on international ERS/ATS guidelines on definition, evaluation and treatment of severe asthma, as well as GINA recommendations.”
Point 3.
The methodology lacks details on the spirometry instruments used, making it difficult to assess the reliability of lung function measurements.
AU: Thank you for your comment. We would like to apologize for not reporting that.
We added the following in methods section as mentioned above:
“More specifically, we performed spirometry in microQuark PC-based spirometer from COSMED, before and after bronchodilator, to assess lung function and seek objective evidence of variable expiratory airflow limitation (>200mL and >12% increase in forced expiratory volume in 1 second).”
Point 4. 
The manuscript does not provide clear criteria for selecting specific biologic therapies for individual patients, introducing a risk of selection bias. Moreover, the inclusion of both allergic and non-allergic severe asthma treatments in the same analysis is methodologically flawed and inappropriate.
AU: Thank you for your comment. We understand your hesitation about coexistence of allergic and non-allergic severe asthma treatments in the same analysis, but in real-life medicine we cannot apply strict inclusion/exclusion criteria and approaches as in randomized controlled trials. It is clearly mentioned in limitations of study that this registry was a prospective study not scheduled to provide mechanistic data and as it was a hospital-based epidemiological study with patient characteristics, quality of data is characterized by sampling bias. With regards to selection of specific biologic therapy, it was based on physicians’ decision taking into account asthma baseline phenotyping, which is presented in table 2.
We added the following phrase in relevant section: “Selection of specific biologic therapy was based on physicians’ decision taking into account asthma baseline phenotyping.”
Point 5. 
Biomarkers such as blood eosinophils and FeNO, although mentioned, are not adequately analyzed in relation to treatment response.
AU: Thank you for your comment. With regards to blood eosinophils, we presented in results that median value of blood eosinophils was lower 6 months after initiation of biologic treatment compared to baseline [365 K/μL (95%CI: 252-448) vs. 100 K/μL (95% CI: 40-121), p<0.0001], a finding that was statistically significant. FeNO values were measured only in baseline, as a consequence, we did not analyze this in relation to treatment response.
Point 6. 
The manuscript states: "7.5% of the study population were administered nebulized ICS plus short-acting beta agonists (SABAs) and short-acting muscarinic antagonists (SAMAs), while 3.8% of patients had not received any inhaled therapy pre-biologic treatment, considering that asthma diagnosis was set following hospitalization due to disease exacerbation." This raises serious concerns about the appropriateness of biologic therapy in patients who were not on high-dose ICS/LABA prior to treatment. According to GINA 2024, biologic therapy is indicated only for severe asthma that remains uncontrolled despite optimized high-dose ICS-LABA treatment or requires high-dose ICS-LABA to maintain control.
AU: Thank you for your comment. We apologize for the misunderstanding. 
Biologic therapy was applied to all patients according to GINA recommendations, as mentioned in methods section. 7.5% of the study population were administered nebulized ICS plus short-acting beta agonists (SABAs) and short-acting muscarinic antagonists (SAMAs), while 3.8% of patients had not received any inhaled therapy pre-triple-inhaled therapy treatment, considering that asthma diagnosis was set following hospitalization due to disease exacerbation.
We modified the section regarding treatment modalities pre biologic agents as follows: “Triple inhaled therapy including inhaled corticosteroids (ICS), long-acting beta agonists (LABAs) and long‐acting muscarinic antagonists (LAMAs) were given to the majority of patients (77.5%, n=62), while 18 patients (22.5%) were given high-dose of ICS/LABA. It is notable that all the aforementioned patients that were given high-dose of ICS/LABA received inhalers contained extra fine particles. Interestingly, 7.5% of the study population were administered nebulized ICS plus short-acting beta agonists (SABAs) and short-acting muscarinic antagonists (SAMAs), while 3.8% of patients had not received any inhaled therapy pre-triple-inhaled therapy treatment, considering that asthma diagnosis was set following hospitalization due to disease exacerbation. Antileukotrienes plus inhalers were implemented in 10 patients (12.5%). Longitudinal use of oral corticosteroids was recorded in 11.3% of included patients.”
Point 7. 
No adverse events are reported, which raises concerns about potential reporting bias or inadequate safety monitoring.
AU: Thank you for your comment. Estimation of adverse events was based on patients’ self-reporting. 
We modified it in text as follows: No treatment related adverse events were noticed based on patients’ self-reporting.
 Point 8. 
The text formatting is inconsistent, particularly between tables and the main body of the manuscript.
The superscript number for affiliations is missing.
The reference list requires review for adherence to journal guidelines, and citation numbers should be placed before the period.
AU: Thank you for your comments. 
The final text formatting including tables was adjusted by journal staff. 
We added superscript number for affiliations.
The reference list was adjusted by journal staff. Citation numbers were placed before the period, as requested.

Round 2

Reviewer 2 Report

Comments and Suggestions for Authors

Dear Authors,

Thank you for submitting the revised version of your manuscript. However, after reviewing the changes, I regret to inform you that the manuscript still presents significant methodological flaws. These issues continue to undermine the validity of the study and the strength of its conclusions.

Additionally, I have identified an inconsistency in the reported data between the original and revised versions of the manuscript. In the first version, at lines 119-122, the authors stated:

"Triple inhaled therapy including inhaled corticosteroids (ICS), long-acting beta agonists (LABAs) and long-acting muscarinic antagonists (LAMAs) were given to the majority of patients (66.2%, n=53), while 18 patients (22.5%) were given high doses of ICS/LABA."

However, in the revised version, at lines 129-132, the text states:

"Triple inhaled therapy including inhaled corticosteroids (ICS), long-acting beta agonists (LABAs) and long-acting muscarinic antagonists (LAMAs) were given to the majority of patients (77.5%, n=62), while 18 patients (22.5%) were given high-dose of ICS/LABA."

The number of patients who received triple therapy increased from 53(66.2%) to 62(77.5%). This discrepancy further raises concerns regarding the accuracy and reliability of the data presented in this study.

Author Response

Thank you for your comments and the fast peer-review process.

Our point-by-point reply to reviewer follows:

Reviewer: 2

Dear Authors,

Thank you for submitting the revised version of your manuscript. However, after reviewing the changes, I regret to inform you that the manuscript still presents significant methodological flaws. These issues continue to undermine the validity of the study and the strength of its conclusions.

Additionally, I have identified an inconsistency in the reported data between the original and revised versions of the manuscript. In the first version, at lines 119-122, the authors stated:

"Triple inhaled therapy including inhaled corticosteroids (ICS), long-acting beta agonists (LABAs) and long-acting muscarinic antagonists (LAMAs) were given to the majority of patients (66.2%, n=53), while 18 patients (22.5%) were given high doses of ICS/LABA."

However, in the revised version, at lines 129-132, the text states:

"Triple inhaled therapy including inhaled corticosteroids (ICS), long-acting beta agonists (LABAs) and long-acting muscarinic antagonists (LAMAs) were given to the majority of patients (77.5%, n=62), while 18 patients (22.5%) were given high-dose of ICS/LABA."

The number of patients who received triple therapy increased from 53(66.2%) to 62(77.5%). This discrepancy further raises concerns regarding the accuracy and reliability of the data presented in this study.

AU:

We thank the reviewer for his/her constructive comments. They helped us to improve the quality of our manuscript. Our point-by-point reply follows:

We have carefully read your comments about our manuscript and we would like to clarify that despite your concerns about methodological limitations, our data clearly presents with the inherent limitations of a prospective observational real-world study. Regarding potential sampling bias the evidence that the percentage of severe asthma patients in our cohort is 5.3% (80/1517) almost identical to that reported in current literature coupled with diagnostic criteria of severe asthma, clearly stated within the manuscript, represent  clear indications that our data presents with minimal, yet acceptable based on the nature of our study, selection bias. 

In addition, with regards to your concerns about clear criteria for selecting specific biologic therapies for individual patients and the inclusion of both allergic and non-allergic severe asthma treatments in the same analysis, we would like to mention that based on International Severe Asthma Registry (ISAR), the global registry for adults with severe asthma -the largest repository of real-world data on severe asthma, curating data on nearly 35,000 patients from 28 countries worldwide- real-world studies can provide complementary data on treatment effectiveness beyond highly-selective RCT patient populations and bridge this “efficacy-effectiveness” gap, providing valuable sources of real-world data on asthma characteristics, trends, and treatment outcomes, which can inform improved management strategies. Importantly ISAR registry clearly revealed that there are overlapping pathogenic pathways based on used biomarkers (IgE, FeNO, BEC) in 65% of enrolled patients. These overlapping endotypes reflect the overlapping phenotypes (allergic eosinophilic, non-allergic eosinophilic, T2 low asthma) that may coexist within the same patient. These data further support our approach to include in our analysis mixed asthma phenotypes, as this is in line not only with real-life clinical cases but also signifies the true heterogeneous nature of the disease per se.   In our cohort, selection of specific biologic therapy was based on physician’s decision taking into account asthma baseline phenotyping (Blood eosinophils, FeNO, IgE, SPTs, comorbidities). Asthma is by definition an heterogeneous disease with overlapping pathogenetic pathways in the majority of patients and we cannot change this reality in an everyday clinical setting.

With regards to the misunderstanding corrected during previous revision, we clearly stated that it was an error.  53 (66.2%) of our patients were initially given triple inhaled therapy, as they had been diagnosed with severe asthma from the beginning. 7.5% of patients were administered SABAs and SAMAs and 3.8% of patients had not received any inhaled therapy pre triple inhaled treatment, considering that asthma diagnosis was set following hospitalization due to disease exacerbation (9 patients in total). These patients were administered triple inhaled therapy before initiating biologic therapy. We corrected this typo in our revised manuscript; yet we do not accept that this typo undermines the clarity and the rigidity of the data presented. For reasons of transparency our data is at your disposal whenever you wish.  

Moreover, with regards to the fact that 18 patients (22.5%) were given high-dose of ICS/LABA without LAMA, our intention is not to communicate the message that physicians should not follow the stepwise approach of GINA recommendations, but in real-life clinical settings physicians in selected cases taking into account severity of previous exacerbations, higher levels of T2 inflammation biomarkers and patient’s preferences can apply biologics in earlier steps (Step 4). To this direction, BRISOTE trial (NCT06750289) which is an ongoing Phase 3b clinical trial assesses the efficacy and safety of benralizumab to reduce exacerbation rate in patients with severe asthma on medium-dose ICS/LABA.

Based on bibliography, 30% of patients with severe asthma do not respond to applied treatment, therefore, we believe that our approach is not an uncommon clinical practice and reflects reality and the real pathophysiology of this heterogeneous disease. In our cases, our approach not to follow the recommendations of GINA for stepwise therapeutic escalations in this minority of patients, was the correct one based on treatment response criteria, as indicated by disease control, elimination of exacerbations even in the absence of OCS and finally based on functional improvement.

Overall, we do not believe that a minor typo undermines the reliability of our data and in order to maintain our clarity and transparency our excel sheet with raw data could be available upon request.

Based on the above and considering that our study presents not only with valuable replicative data but also with novel evidence, as this is the first real-world study showing effectiveness of Tezepelumab in a Greek population of severe asthmatic patients , we truly believe that our manuscript could make a good fit in the journal.

Sincerely,

Argyrios Tzouvelekis MD, MSc, PhD

Professor of Respiratory Medicine

Head Department of Respiratory Medicine

University of Patras, Greece

Associate Professor Adjunct, PCCSM, Yale School of Medicine, USA

atzouvelekis@upatras.gr, argyris.tzouvelekis@gmail.com
